# Soy Protein/Polyvinyl-Alcohol (PVA)-Based Packaging Films Reinforced by Nano-TiO_2_

**DOI:** 10.3390/polym15071764

**Published:** 2023-04-01

**Authors:** Xueying Tian, Zhizhou Chen, Xiaomeng Lu, Jianlou Mu, Qianyun Ma, Xiaoyuan Li

**Affiliations:** 1College of Food Science and Technology, Hebei Agricultural University, Baoding 071000, China; 2College of Mechanical and Electrical Engineering, Hebei Agricultural University, Baoding 071000, China

**Keywords:** soy protein-based biodegradable film, PVA, nano-TiO_2_

## Abstract

This work was investigated to prepare a reinforcing composite packaging film composited of soy protein/polyvinyl alcohol (PVA) and nano-TiO_2_. First, different film compositions were designed by the particle size of nano-TiO_2_, concentration of nano-TiO_2_, concentration of polyvinylpyrrolidone (PVP, a dispersing agent for nano-TiO_2_), and pH of film casting solution. Then, the film composition that yielded the optimal physical properties was identified using orthogonal array design single-factor experiments, considering its physical properties, including tensile strength, elongation, water absorption, water vapor transmission, oxygen permeation, thermal property, and film morphology. The results displayed that the optimal film composition was (1) soy protein/PVA film with 2.5 wt% nano-TiO_2_, (2) 30 nm nano-TiO_2_ particle size, (3) 1.5 wt% PVP, and (4) pH 6.0 of film-forming solution. It yielded tensile strength of 6.77 MPa, elongation at break rate of 58.91%, and water absorption of 44.89%. Last, the films were characterized by scanning electron microscope (SEM) and differential scanning calorimetry (DSC). SEM analysis showed that compared with the film without TiO_2_, the film containing TiO_2_ has a smoother surface, and DSC determined that adding nano-TiO_2_ can improve the thermostability of soy protein/PVA film. Therefore, the film prepared in this paper is expected to provide a new theoretical basis for use in the packaging industry.

## 1. Introduction

Traditional petroleum-based plastic materials are widely used due to their properties and cost-effectiveness; however, they also possess negative environmental effects created during decomposition and the limitation of petroleum resources [1]. The traditional plastic film preparation process consumes a lot of energy, and the added substances in the process of use migrate to the packaged food. After use, it is difficult to reuse and deal with it, which causes great trouble to nature. At the same time, after the erosion of the waste plastics for years by wind and rain, the harmful substances in the plastic products penetrate the underground, reducing the land quality as agricultural farmland and reducing a large number of crops, posing a threat to the grain industry. In addition, although the energy generated by the incineration of waste plastics can be used to generate electricity, a large amount of harmful gases previously added to the plastic products during the incineration are released. Tremendous efforts have been spent to develop bio-based materials, and a few potential candidates such as cellulose-, starch- [2], and protein [3]-based materials have been identified. Soybean planting land is extensive, and the soybean crop contains more nutrients which are beneficial to the human body. Soybean has the highest protein quality score, about 40%, making it a high protein crop. Among them, soy protein has been investigated extensively because of its natural abundance [4], low cost, and biodegradability [5]. In addition, its secondary, tertiary, and quaternary levels of structures, and various reactive side groups such as -NH2, -OH, and -SH, result in different possibilities of binding and crosslinking reactions; therefore, different methods can be used to modify its chemical and physical properties. For example, Hernandez-Izquierdo et al. [6] studied the role of thermoplastic processing , Ou et al. [7] added ferulic acid to film-forming solutions, and Kumar et al. [8] and Tang et al. [9] added aligned ramie fiber and transglutaminase to reinforce the structure of soy protein film.

Soy protein-based packaging film has a wide range of raw materials and can be completely degraded. It can reduce the pollution caused to the environment in the process of use, alleviate the current situation of national energy shortage, and make contributions to the sustainable development of China. Usually, the method of soy protein film selection is obtained. The premise of soy protein film formation is to deform it by heating, acid, or alkali, and the disulfide bond between the molecules is broken and the protein is dispersed to form a solid structure. At this time, the hydrophilic group exists in the water, and a stable protein layer is formed in the water, thus obtaining the soy protein film. The evaluation criteria of soy protein film are usually thickness, tensile strength, break elongation rate, and barrier properties.

Nanoparticles were also explored to improve the mechanical strength of soy protein film [10]. Such nanoparticles include supramolecular nanoagregates, hydroxylpropyl lignin [11,12], and layered silicates [13]. Particularly, TiO_2_ is non-toxic and tasteless, and nano-TiO_2_ has the size of the nanoscale, equivalent to about 1/10 of the TiO_2_ size, so it is also called ultra-fine TiO_2_. Anatase TiO_2_ has a tetragonal crystal structure [14], which easily forms a regular octahedral shape with thermodynamic stability. Compared to anatase, rutile TiO_2_ has a tetragonal crystal structure, and it has a smaller bandgap, a higher dielectric constant, and a higher refractive index, but rutile films are made at temperatures that are too high. Therefore, it is generally applied to the temperature-sensitive packaging materials [15]. Brookite TiO_2_ has an orthorhombic crystal structure. Brookite TiO_2_ is the rarest phase in the TiO_2_ polycrystalline type [16].

Nano-TiO_2_ is a new photocatalyst antibacterial agent [17] which can produce the strong oxidative active substances ·OH and ·OOH and then destroy the organic molecules to kill microorganisms. Studies showed that the addition of nano-TiO_2_ to soybean separation protein membrane results in antibacterial enhancement and evenly uniform nano-TiO_2_ in soy protein film liquid. Through the modified film’s mechanical properties, permeability and infrared spectrum analysis found that the interactions between nano-TiO_2_ and soybean protein nanoparticles are evenly dispersed to the soybean protein peptide chain side chain, making the previous film closer to the main membrane material soybean protein structure; the microscopic performance for crystallinity and the macroscopic performance improve the mechanical properties. However, the addition of nano-TiO_2_ reduces the flexibility of the films [18]. Studies have shown that nanoparticles with a small diameter film reach a maximum tensile strength. Larger diameter particles affect the formation of the film, making the film fragile [19]. The crystallinity of TiO_2_ nanoparticles seems to be affected by Fe_3_O_4_ [20], acting as nano-seeds to improve the tetragonal TiO_2_ anatase structure with respect to the amorphous one when the TiO_2_ nanoparticles have a small size with spherical morphology. The smaller the particle size, the larger the active site and specific surface area [21].

In the field of modified non-synthetic polymer composite membranes, in addition to the plant protein packaging membrane, nano-TiO_2_-modified chitosan membranes and cellulose membranes are also very common. Chitosan is a natural material with a high molecular weight. Due to its molecular weight of up to 50,000, its cell surface usually forms a polymer formation membrane, which can prevent the invasion of bacteria and viruses; that is to say, chitosan itself is antibacterial. Cellulose has many advantages, including biodegradability and low cost. It is very consistent for the needs of current environmental development. Cellulose membrane can improve the hydrophilic energy and mechanical strength of the membrane [22].

PVA is a semicrystalline synthetic biopolymer that is easily miscible in water [23]. Each of its polymeric units are linked together via C–C bonds. It is chemically prepared by suspension polymerization of poly (vinyl acetate). It is broadly used as hydrogel in drug delivery and tissue engineering applications [24]. It improves the film-forming ability of pristine SPI. Various nanoparticles or nanofibers are doped with PVA for enhancing mechanical properties, or it is used in packaging or wound dressing applications [25]. PVA film has high light transmittance, high mechanical strength, printing properties and oil resistance and belongs to water-soluble films. However, PVA film also has some disadvantages. Due to its good water solubility, PVA is limited when making it into packaging film alone, so it needs to clamp PVA between two layers of hydrophobic materials such as PE, which not only overcomes the hydrophilic defects of PVA but also uses the advantages of high PVA with mechanical strength and high light transmittance.

Many studies have shown that soy-protein-based packaging film, nano-TiO_2_ particles, and poly vinyl alcohol (PVA) have been added individually to soy protein to form a more practical and durable film [26]. For instance, TiO_2_ was added in packaging film or packaging coating to improve the mechanical strength due to the crosslink between TiO_2_ and soy protein, and such film also showed antimicrobial activity against food-borne pathogens [27]. PVA is a synthetic biodegradable material which has been widely used in food and other industries. Adding PVA to soy protein can also improve the mechanical strength of the composite film because of the interactions between soy protein and PVA through hydrogen bonding, dipole–dipole interactions, and enhanced physical entanglement. Xiaorong Liu et al. [28] prepared composite film via the casting method. The results showed that the addition of PVA improved the mechanical response and thermal properties of the matrix of soybean protein and did not affect its transparency so that the film had a more stable structure.

Although the performance of soy protein mixed with PVA has been greatly improved, it is far from the plastic widely used plastic in modern times. It is worthwhile to explore the effects of their combination as they have different crosslinking reactions with soy protein due to TiO_2_ and PVA having shown their individual effects to improve the mechanical strength of soy protein film. Therefore, we need film with better air permeability, more tensile resistance, and more stable structure. The objective of this paper was to evaluate the effect of the film composition of soy protein/PVA-based and nano-TiO_2_-reinforced composite film on its physical properties. Only by thoroughly mastering the performance analysis and mechanism of soy protein/PVA-based packaging films reinforced with nano-TiO_2_ can we get closer and closer to replacing the position of plastic products.

## 2. Materials and Methods

### 2.1. Materials

Soy protein was purchased from Anyang Mantianxue Food Manufacturing Co., Ltd. (Baoding, Hebei, China) with protein concentration of 60% (Kjeldahl method). PVA (1750 ± 50) was provided by Kermel, Tianjin Chemical Reagent Co., Ltd. (Tianjin, China). Nano-TiO_2_ (15 nm, 30 nm, 50 nm) was purchased from Xuancheng Crystal Sharp New Materials Co., Ltd. (Xuancheng, Anhui, China). Polyvinylpyrrolidone (PVP, K-30, average molecular weight 30,000) was purchased from Sigma-Aldrich (St. Louis, MO, USA). Glycerol (99.9%) was purchased from Tianjin Tianda Chemical Experiment Plant (Tianjin, China).

### 2.2. Film Composition

Four single factors were studied: (1) particle size of nano-TiO_2_—15, 30, and 50 nm; (2) concentration of nano-TiO_2_—0, 0.5, 1.0, 1.5, 2.0, 2.5, 3.0, and 3.5%; (3) concentration of PVP (dispersing agent for nano-TiO_2_)—0, 0.5, 1.0, 1.5, 2.0, 2.5, 3.0, and 3.5%; and (4) pH of casting solution—5, 6, 7, 8, 9, 10, and 11. According to the results from the single-factor experiments, a standard L9(3^4^) orthogonal array design experiment was conducted to identify the optimal film composition.

### 2.3. Film Preparation

I referred to the method of Hasfalina Che Man et al. [29] with modifications. Nano-TiO_2_ and PVA were mixed in distilled water and further stirred at ambient temperature for 1 h to form a PVA/nano-TiO_2_ suspension. The suspension was kept and used as stock solution. Soy protein was dissolved in distilled water at 70 °C for 30 min, after which PVA/nano-TiO_2_ suspension were added and the pH of the solution was adjusted. The solution was then mixed at 90 °C for 10 min, and glycerol (99.9%) was added and followed by another 30 min of mixing at 90 °C. The mass fraction of glycerol was 2.0%. Glycerol is a plasticizer that can improve elongation at the break of the film. Vacuum filtration method was then used to filter out the non-dissolved particles. The collected solution was then coated on a glass plate to dry at 90 °C. The coated plate was kept at room temperature for 24 h before peeling off the film.

### 2.4. Physical Properties Evaluation

Tensile strength and elongation of film samples, 150 mm × 15 mm, were measured by a XLW (PC) Intelligent Electronic Tensile Testing Machine (The Labthink Technology Co., Ltd. Jinan, Shandong, China). The experiment was repeated three times, and the average value was obtained.

The tensile strength and fracture elongation are calculated as follows.
M = F × 10^−6^/S(1)
where M is the tensile strength of the sample in MPa; F is the maximum tensile force of the sample during complete fracture in N; S is the cross-sectional area of the sample in m^2^.
E = (L_1_−L_0_)/L_0_ × 100%(2)
where E is the elongation of sample fracture; L_1_ is the length of the film in m; L_0_ is the original length of the film in m.

### 2.5. Water Absorption

Film sample (50 mm × 12 mm) was dried in an oven until no weight loss was observed, and the weight was recorded (W_0_). It was then soaked in 300 mL of water for 24 h. After removing from water, tissue paper was used to wipe the water on the surface of the film, and the film weight was recorded (W_1_). Water absorption (%) was calculated using (W_1_ − W_0_)/W_0_ × 100. The experiment was repeated three times, and the average value was obtained.

### 2.6. Comprehensive Film Property Evaluation

A mathematical function X(u) was used to quantitatively assign a grade to each property to indicate the effect of each factor on the film property. The calculation of X(u) of each film property is as follows.
X(u) = (Xi − Xmin)/(Xmax − Xmin) (positive effect including elongation and tensile strength)(3)
X(u) = 1 − (Xi − Xmin)/(Xmax − Xmin) (negative effect, water absorption)(4)
where Xi is the measured value of film property, Xmax is the maximum measured value, and Xmin is the minimum measured value of the film.

The comprehensive property was calculated by adding X(u) for each film property based on the designated proportion of each property. The proportion of tensile strength was 40%, elongation was 30%, water absorption was 20%, and visual considerations were 10%. The assignment of higher proportion for tensile strength and elongation is because bio-based polymers generally do not possess good physical properties for practical application, and improvement in physical properties is critical for evaluation. Water absorption also has a high proportion in evaluation because soy protein is water soluble, and the modification has to improve the water resistance of the film.

### 2.7. Water Vapor Transmission and O_2_ Permeation

Water vapor transmission rate and O_2_ permeability of the film (composition identified by orthogonal array design) with varying concentrations of nano-TiO_2_ of 0.25%, 0.5%, 1.0%, 1.5%, 2.0%, and 2.5% were tested and compared to commercially available synthetic films to evaluate their potential application. Water vapor transmission was measured using a W3/030 water vapor transmission analyzer (Labthink Instruments Co. Ltd., Jinan, China) and O_2_ permeation was measured using a VAC-VBS gas permeation analyzer (Labthink Instruments Co. Ltd., Jinan, China). The experiment was repeated three times, and the average value was obtained.

### 2.8. Thermal Properties

Thermal properties were measured by differential scanning calorimetry (DSC) analysis using a DSC-200PC machine (NETZSCH, Selb, Free State of Bavaria, Germany). Approximately 10 mg of sample was placed in hermetically sealed aluminum pans. The sample was heated from room temperature to 300 °C at rate of 10 °C/min under N_2_ environment with gas flow of 100 mL/min.

### 2.9. Visualization of Film Structure

Film strips were examined using JSM-7001F Schottky Emission Scanning Electron Microscope (JEOL, Akishima, Japan) with 15 kV voltage.

## 3. Results

### 3.1. Effects of Particle Size of Nano-TiO_2_ on the Film Properties

Film composed of 1.5% nano-TiO_2_ and 1.0% PVP was used for evaluation. Table 1 shows that the optimal particle size tested was 30 nm, which was selected for further evaluations. Increasing it from 15 nm to 30 nm resulted in higher tensile strength and elongation and lower water absorption, but further increasing it to 50 nm resulted in reduced film tensile strength and elongation and higher water absorption. The apparent increase in the elongation at break of the film from 15 nm to 30 nm may be due to the addition of glycerol also increasing the elongation at break of the film. Studies have shown that a smaller particle size rapidly increases the specific surface area of nanoparticles in more active sites on the surface, which enhances the performance of the membrane [30].

### 3.2. Effect of Nano-TiO_2_ Concentration

Figure 1 and Figure 2 show that 2.5% nano-TiO_2_ was the optimal concentration to yield the best film properties with the highest tensile strength and elongation at break and lowest water absorption. As the concentration of TiO_2_ increased, the film tensile strength and elongation at break increased first and then decreased. The reason was similar to the calculation results of the particle size of nano-TiO_2_. At an appropriate concentration, there were more active sites. When the concentration was too low, the specific surface area of nanoparticles was too small, and there were fewer active sites. When the concentration was too high, the specific surface area of nanoparticles was too large, and there were fewer active sites, leading to poor film performance. In addition, the threaded structure of soy protein greatly improved the tensile strength of the film [31].

### 3.3. Effect of PVP Concentration

Film samples containing 2.5% nano-TiO_2_ were used for evaluation. The concentration of PVP can appropriately enhance the tensile strength of the film samples [32]. Results (Figure 3 and Figure 4) showed that when PVP concentration was 1 %, tensile strength reached the highest value, water absorption reached the lowest value, and elongation at break was close to the highest value. The reason behind this was that when the amount of PVP used was small, the size of nano-TiO_2_ particles was larger. If the concentration of PVP was too high, excessive PVP made the nano-TiO_2_ particles smaller and spherical. Therefore, the strength and toughness of the film were also affected [33].

### 3.4. Effects of pH of Film Casting Solution

Film samples containing 2.5% nano-TiO_2_ and 1.0% PVP were used for evaluation. As shown in Figure 5 and Figure 6, when the pH was 7.0, both film tensile strength and elongation at break reached the maximum, and water absorption was the lowest. Therefore, pH 7.0 was the optimal. In addition, pH also had an effect on the visual property of the film. When pH was 5.0, there was white precipitation observed, while when pH was higher than 10.0, the color of the film turned yellowish. Soy protein has pI of 4.5, at which point positive and negative charges are balanced in soy protein solution and precipitation occurs, and when pH is below or above pI, soy protein dissolves [34]. Our results showed that precipitation occurred at pH around 5.0, indicating that the interaction between soy protein, PVA, and nano-TiO_2_ altered charge distribution of the system and the pI of soy protein. At optimal pH of 7.0, soy protein bore negative charges, and the molecular repulsion led to the optimal physical property. The pH change (Zhao et al., 2023 [35]) led to the protonation and deprotonation of protein, which affected the charge properties and densities of the material surface. The redistribution of attraction and repulsion between soybean proteins led to the pH change of coating solution, which occurred to the side-chain ionization rate of soybean protein.

### 3.5. Orthogonal Array Design Experiment

According to the single-factor experiment results shown above, a standard L9(3^4^) orthogonal array design was conducted to optimize the formulation of the film, and Table 2 shows the results obtained.

The optimum condition was A_2_B_3_C_1_, which contains 2.5 wt% nano-TiO_2_ (30 nm) and 1.5 wt% PVP in the film, and the processing pH is 6.0.In the optimized film A_2_B_3_C_1_, the most important factor that affects the film properties should be PVP concentration, followed by nano-TiO_2_ (30 nm) concentration and pH.

This result further verified the addition of PVP and nano-TiO_2_, controlling the pH of the casting solution and enhancing the performance of the film. Nano-TiO_2_ increased the active sites of the film and enhanced the tensile resistance of the film [30]. The appropriate PVP concentration changed the particle size of nano-TiO_2_, thereby affecting the performance of the film. The appropriate pH changed the interaction between soy protein, PVA, and nano-TiO_2_, thereby making the film structure more stable [35].

### 3.6. Validation of the Optimum Condition

Experiments were conducted to validate the properties of A_2_B_3_C_1_. Film was made using the same method mentioned previously, and the experiment results are shown in Table 3.

Table 3 shows that the experimental results were consistent with results obtained from orthogonal array design experiments. Compared to some soy-protein-based films modified by other methods reported in the literature, A_2_B_3_C_1_ has equivalent or superior mechanical properties [36,37,38,39]. Compared to synthetic polymers in general, A_2_B_3_C_1_ film still has lower tensile strength. The only comparable synthetic materials were low-density polyethylene (LDPE) and polytetrafluoroethylene (PTFE), whose tensile strength ranges from 9.65 to 20.67 MPa. This result confirmed the previous statement that although the tensile strength of the film was relatively low, it has been improved by the addition of nano-TiO_2_ and PVP. The addition of nano-TiO_2_ and PVP also improved the elongation and water absorption of the film.

### 3.7. Effects of Nano-TiO_2_ on Water Vapor Transmission Rate and O_2_ Permeation

A series of nano-TiO_2_ concentrations of 0.25%, 0.5%, 1.0%, 1.5%, 2.0%, and 2.5% were evaluated for their effects on the water vapor transmission rate and O_2_ permeability. Results (Figure 7) showed that as the nano-TiO_2_ concentration increased, water vapor transmission and O_2_ permeation decreased to minimum at 3607.27 g m^−2^ 24 h^−1^ MPa^−1^ (nano-TiO_2_ concentration 1.5%, water vapor permeability 46.89 g mil 100^−1^ in^−2^ day^−1^ atm^−1^) and 200.68 cm^3^ m^−2^ 24 h^−1^ 0.1 MPa^−1^ (nano-TiO_2_ concentration 1.0%, O_2_ permeability 25.8 cm^3^ mil 100^−1^ in^−2^ day^−1^ atm^−1^), respectively. Further increasing nano-TiO_2_ concentration caused increase in both water vapor transmission and O_2_ permeation. The reason behind this the incorporation of nano-TiO_2_ improved the oxygen resistance of the film and impeded water permeation through the film. The addition of nano-TiO_2_ improved the water resistance of the film. With the addition of titanium dioxide, more orderly and dense crystalline regions were formed in the film, resulting in a decrease in the water holding capacity of the film. In addition, hydrophilic properties affected the transport performance of oxygen and water vapor. TiO_2_ was hydrophilic, and when the concentration was too high, its water solubility in the polymer increased to assist in the transport of water vapor and oxygen [40]. Observation was that when TiO_2_ was added initially, the structure of the film became more packed due to intermolecular electrostatic force and crosslinking, making water vapor and O_2_ difficult to permeate through the film [41]. The crosslinking between nano-TiO_2_ and soy protein also forms a tortuous path to hinder the permeation of water vapor and O_2_ [42]. In addition, TiO_2_ is hydrophilic, and when the concentration is too high, water solubility in the polymer increases to help water vapor transmission. Compared to synthetic polymers, the results were comparable to the O_2_ permeability of plasticized polyvinyl chloride (PVC), which ranges from 50 to 1500 cm^3^ m^−2^ 24 h^−1^ 0.1 MPa^−1^, and were slightly higher in water vapor permeability than PVC (15–40 g mil 100^−1^ in^−2^ day^−1^ atm^−1^). It indicates that the soy protein composite film could be a potential replacement for PVC given the concern of its chloride component on human health [43].

### 3.8. Effect of Nano-TiO_2_ on Thermal Properties of Soy Protein/PVA Film

DSC provided a more rapid and obvious method for experiments. DSC accurately controlled the temperature in a short period of time and controlled the temperature range. The DSC provided a convenience for large-scale experiments [44].

DSC results (Figure 8) showed that adding nano-TiO_2_ can improve the thermostability of soy protein/PVA film. In the temperature range of 20–30 °C, film without TiO_2_ had a dramatic change in heat adsorption; however, film with TiO_2_ did not change as dramatically. In the process of temperature change, the heat absorbed by soybean protein/PVA was greater than that absorbed by the composite film modified by nano-TiO2, which indicated that the addition of nano-TiO2 restrains the movement of the polymer in the original composition, which verified that the modified film structure was more stable. The results showed that after adding the nano-TiO_2_, intermolecular interactions between nanoparticles reduced the flexibility of molecular chains [45]. The addition of nano-TiO2 increased the crystallinity and impeded the movement of molecular chains. Thus, the composite film increased its melting temperature and made its structure more stable. It was also possible that glycerol played a plasticizing role, making its structure more stable and resulting in better thermal stability.

### 3.9. Effect of Nano-TiO_2_ on Soy Protein/PVA Film Morphology

Figure 9 shows that adding TiO_2_ improved the smoothness of the polymer structure. In Figure 9A, although soy protein and PVA were mixable, the film morphology was rough, as shown by the net-like structure. Adding TiO_2_ (Figure 9B) made the structure more packed, and the added TiO_2_ was uniformly distributed throughout the polymer structure without aggregation. After adding nano-TiO_2_ to the film, the mechanical properties and barrier properties of the film were significantly improved [46]. The results correlated with DSC results because a more packed polymer structure has better thermostability [47].

## 4. Conclusions

In this paper, soy protein/PVA/nano-TiO_2_ film with excellent physical properties was successfully prepared. The addition of nano-TiO_2_ and its dispersing agent PVP can reinforce soy protein/PVA film structure through electrostatic interaction and crosslinking, but at higher concentrations, it can negatively impact the film structure due to the coagulation of nano-TiO_2_. Adjusting pH can also modify film structure by electrostatic interaction, and addition of nano-TiO_2_ changed the pI of soy protein. The optimum film property was obtained with film formulation of 2.5% nano-TiO_2_ (30 nm), 1.5% PVP, and pH 6.0. It is more environmentally friendly than the traditional plastic film. Compared to the bionic soy protein film, the film obtained by our research has stronger stability and better tensile properties. Other aspects of performance have also improved. The resulting film yielded mechanical properties equivalent or superior to the soy protein films modified by various other methods reported in the literature and were close to two synthetic polymers, LDPE and PTFE. Adding nano-TiO_2_ can also yield film with water vapor transmission rate and O_2_ permeability comparable to PVC. Therefore, the prepared biocomposites are low-cost material and used as a packaging material due to ecofriendly behavior of soy protein/PVA/nano-TiO_2_. This material conforms to the modern concept of green development; it is conducive to environmental protection, and it has considerable development prospects.

## Figures and Tables

**Figure 1 polymers-15-01764-f001:**
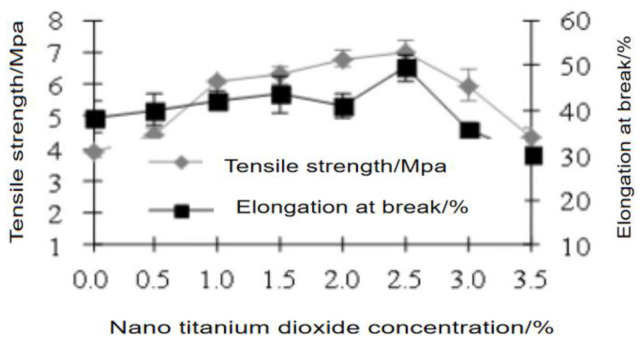
Effects of nano-TiO_2_ concentration on tensile strength and elongation rate of the film.

**Figure 2 polymers-15-01764-f002:**
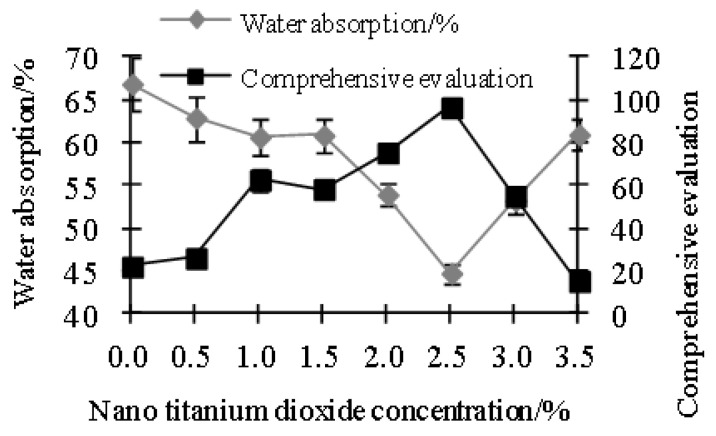
Effects of nano-TiO_2_ concentration on water absorption and comprehensive evaluation.

**Figure 3 polymers-15-01764-f003:**
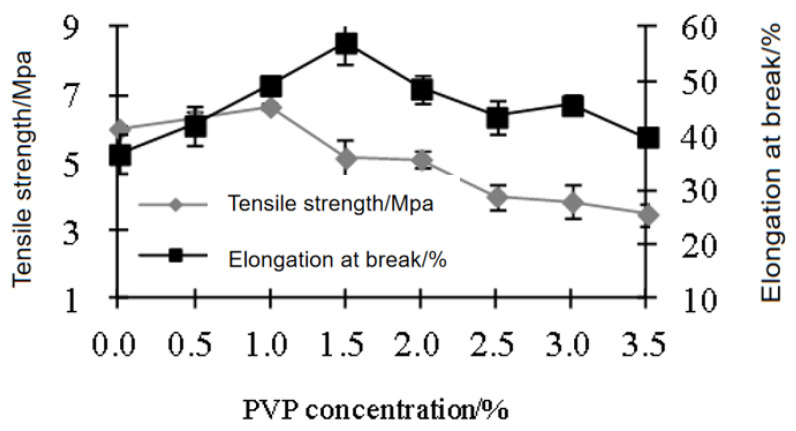
Effects of PVP concentration on tensile strength and elongation rate.

**Figure 4 polymers-15-01764-f004:**
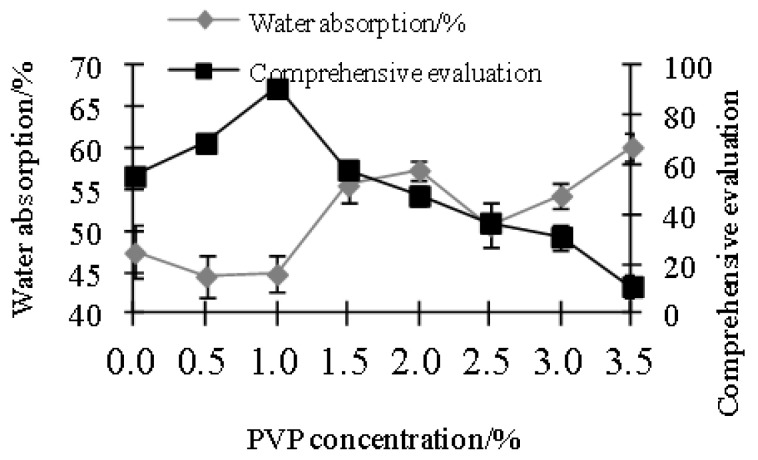
Effects of PVP concentration on water absorption and comprehensive evaluation.

**Figure 5 polymers-15-01764-f005:**
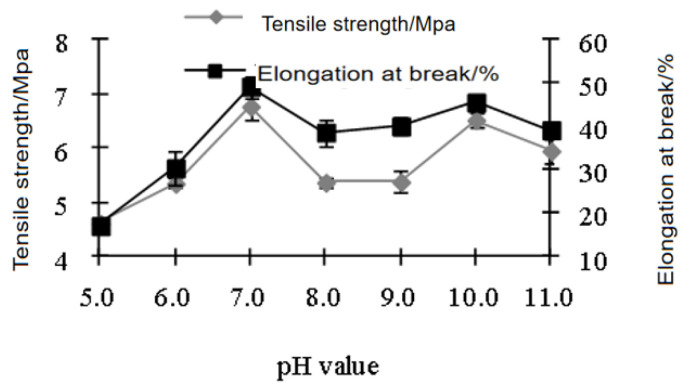
Effects of pH on tensile strength and elongation rate.

**Figure 6 polymers-15-01764-f006:**
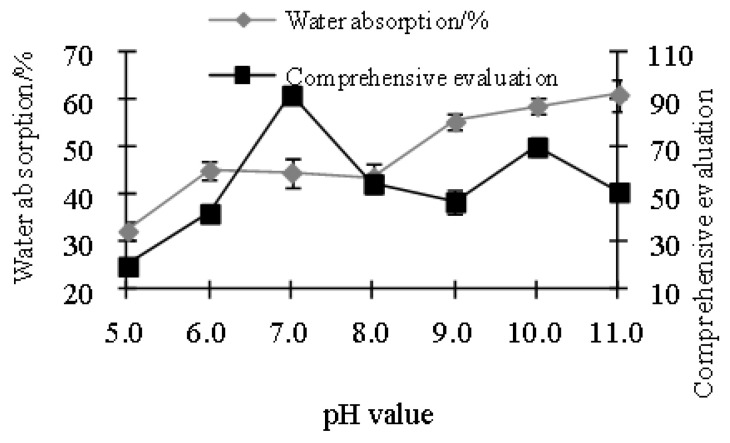
Effects of pH on water absorption and comprehensive evaluation.

**Figure 7 polymers-15-01764-f007:**
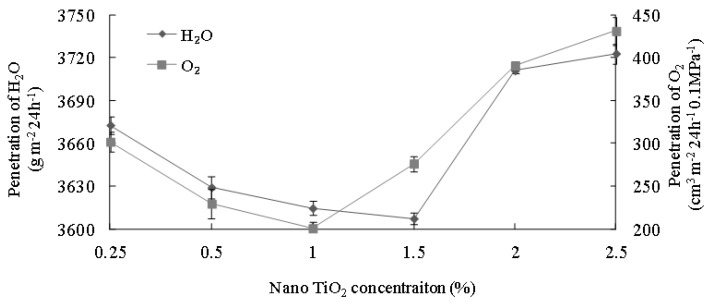
Effects of Nano-TiO_2_ concentration on water vapor transmission and O_2_ permeation.

**Figure 8 polymers-15-01764-f008:**
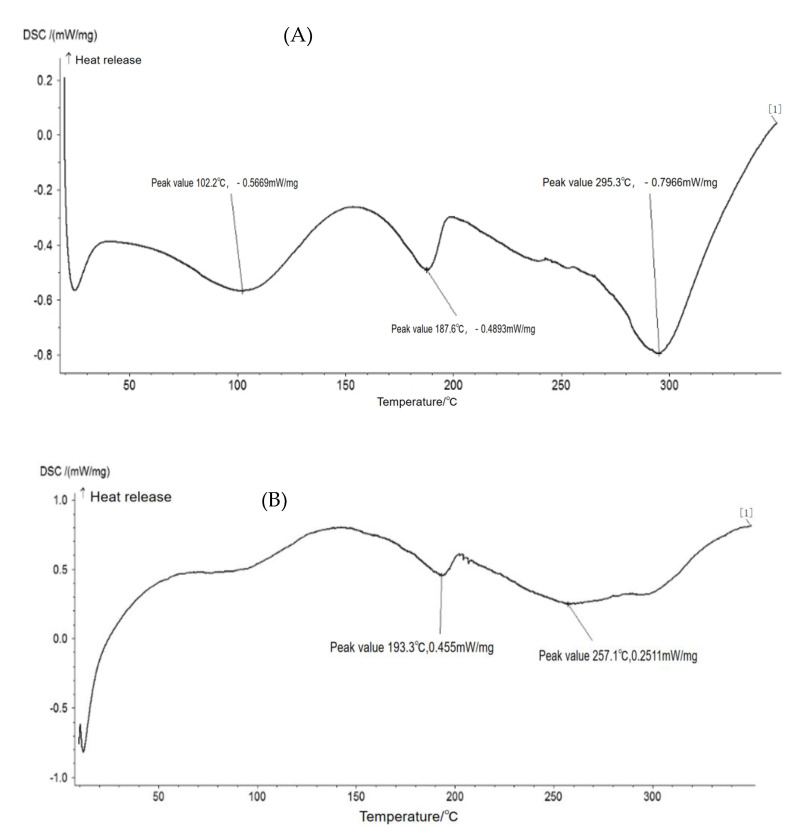
DSC results of soy protein/ polyvinyl alcohol composite film ((**A**) without TiO_2_, (**B**) with TiO_2_).

**Figure 9 polymers-15-01764-f009:**
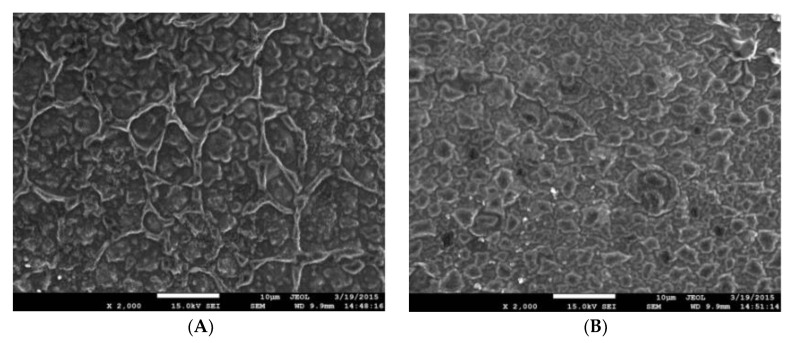
SEM images of soy protein/polyvinyl alcohol composite film ((**A**) without TiO_2_, (**B**) with TiO_2_) (material composition: 3.0% soy protein, 3.0% PVA, 2.0% glycerol, 1.5% PVP, 2.5% nano-TiO_2_).

**Table 1 polymers-15-01764-t001:** Effects of particle size of nano-TiO_2_ on the properties of the film.

Particle Size	Tensile Strength (MPa)	Elongation (%)	Water Absorption (%)	Comprehensive Evaluation
15 nm	4.08 ± 0.45 ^b^	38.64 ± 2.98 ^b^	59.22 ± 1.98 ^a^	15.52 ± 2.31 ^c^
30 nm	5.69 ± 0.53 ^a^	53.40 ± 2.80 ^a^	45.00 ± 2.01 ^c^	90.00 ± 0.87 ^a^
50 nm	4.63 ± 0.54 ^b^	51.20 ± 3.62 ^a^	54.71 ± 4.32 ^ab^	50.10 ± 1.03 ^b^
Control	3.85 ± 0.18 ^b^	38.38 ± 2.09 ^b^	52.77 ± 2.36 ^b^	12.40 ± 2.43 ^c^

Mean values of 6 replicates ± standard deviations are given.Different superscript letters (^a^, ^b^, ^c^, ^ab^) in the same column indicate significant differences (*p* < 0.05).

**Table 2 polymers-15-01764-t002:** Results and analysis of the filming orthogonal array design experiments.

Experimental Number	Factor				Comprehensive Evaluation
	Nano-TiO_2_	PVP	pH	Error	
	A [%]	B [%]	C	D	
1	1 (2.0%)	1 (0.5%)	1 (6)	1	38.15
2	1	2 (1.0%)	2 (7)	2	41.01
3	1	3 (1.5%)	3 (8)	3	43.99
4	2 (2.5%)	1	2	3	41.36
5	2	2	3	1	45.41
6	2	3	1	2	88.60
7	3 (3.0%)	1	3	2	13.14
8	3	2	1	3	37.66
9	3	3	2	1	59.34
T1	123.15	92.65	164.41	142.90	
T2	175.38	124.08	141.72	142.76	
T3	110.15	191.94	102.54	123.02	
R1	41.05	30.88	54.80	47.63	
R2	58.46	41.36	47.24	47.59	
R3	36.72	63.98	34.18	41.01	
M	21.74	33.10	20.62	6.58	

**Table 3 polymers-15-01764-t003:** The validation results of the A_2_B_3_C_1_.

Combination	Tensile Strength(MPa)	Elongation (%)	Water Absorption (%)	Comprehensive Evaluation
A_2_B_3_C_1_	6.77 ± 0.10	58.91 ± 3.12	44.89 ± 2.85	88.79 ± 1.45

## Data Availability

Not applicable.

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
