# Peer review of "Soy Protein/Polyvinyl-Alcohol (PVA)-Based Packaging Films Reinforced by Nano-TiO2"

_polymers, 2023, doi:10.3390/polym15071764_

Round 1

Reviewer 1 Report

1. There are many grammar and spelling mistakes in the manuscript. 

2. Introduction part needs some revision. A more comprehensive literature review should be included. For example, excluding TiO2, what are the other types of NPs that have been used in soy protein films?  What is the current research stage of TiO2 in soy protein films? what are the advantages and disadvantages? Will the size, shape, crystal structure of TiO2 NPs affect their performances?

The authors also mentioned that PVA has been added individually into the soy protein but there is no corresponding reference included. 

3. Some typical stress-strain curves should be included in the main text or in the supporting information to support Figures 1-6. 

4. Results should be discussed in more detail.

For example, lines 155-157 "The results show that as particle size increased, distance among individual particles became smaller and inter-particle electrostatic forces became stronger" I suggest the authors to conduct some TEM experiments to confirm this statement. 

lines 279-280 "In the temperature range of 20-30°C, film without TiO2 had a dramatic change in heat adsorption, however, film with TiO2 did not change as dramatic." Instead of "dramatic change", more quantitative analysis is needed. 

5. Mechanism should be explained in more detail and more clearly. I suggest the authors to add a scheme or figure to show the structures and interactions between soy protein, PVA and TiO2 to help the readers understand the concept.

6. What is the shape and crystal structure of TiO2 NPs in the current study? this should be given in the main manuscript

Author Response

I have uploaded the reply in file form》

Reviewer 2 Report

The manuscript polymers-2261812 deals with a generally interesting scientific and technical topic. However, it does not follow a logical flow of argumentation and needs major revision.

In the title, it is stated that the composites are intended to be (food?) packaging films.

So, the introduction should explicitly specify the characteristics needed for this type of application: mechanical properties (which ones in particular?), barrier properties (O2, water vapors...), risk of migration of additives, etc., thus explaining the type of measurements to be performed on the composites under study.

Secondly, the soy protein/PVA blends for packaging films are largely presented in the literature. An overview of the typical processing methods, range of compositions and main characteristics obtained so far, should be briefly presented in the Introduction part.

Thirdly, the addition of different types of nanoparticles (NPs) to soy protein/PVA blends has been already reported in the literature and the main effects of their chemical nature, size and shape on the reinforced soy protein/PVA films should also be briefly presented in the Introduction part, as a starting point for the present work.

Finally, the present choice of TiO2 nanoparticles for soy protein/PVA films, vs. other types of nanoparticles (already reported in the literature) should be clearly explained.

In the Experimental part:   Overall, the preparation and characterization methods are correctly chosen and described.

In the Results part:

-  A general table with all the samples prepared + their exact composition is necessary (Soy protein / PVA / TiO2 NPs / PVP)

-       In §3.1, §3.2, §3.3 and §3.7, the physical explanations of the experimental results are not fully consistent, and the cited literature is not correctly exploited.

In addition, the effect of the particle size and concentration of TIO2 NPs, as well as the effect the PVP content (as NPs dispersing agent) should be based on SEM images for each type of film under study. Or, this type of direct information is entirely missing, making it impossible to investigate the dispersion quality of NPs into the Soy protein / PVA matrix, the presence or not of NPs agglomerates, the possible interactions etc.

-       In §3.8, the thermal behavior of the films at 20-30°C is not physically explained. In addition, the manuscript details the thermal behavior of the films in the range from 180°-195°C, without specifying the practical interest of this range of temperatures.

-       The section §3.9 (SEM study) should be presented at the very beginning of the Results part, and should be extended to all the compositions studied in this work, in order to provide direct information on the different obtained morphologies, and to support clear physical interpretation of the experimental results from parts §3.1, §3.2, §3.3 and §3.7.

-       Following a logical presentation flow, the parts §3.5 and 3.6 should figure after the different paragraphs that investigate and explain the physical influence of different factors affecting the final properties of the film packaging (i.e., size and concentration of NPs, PVP content, etc.)

-       Conclusions should be reformulated, with respect to more consistent physical explanations of the experimental results.

In addition, lines 296-297 mention: In this paper, soy protein/PVA/nano-TiO2 film with excellent physical properties have been successfully prepared.” Or, this general conclusion is not obvious, from this version of the manuscript. It appears necessary to explain why the obtained results are excellent – by comparing them with literature data on classical packaging films and/or at least with comparable soy protein/PVA/NPs films reported in the literature.

Considering all these elements, I recommend a major revision of the manuscript.

Author Response

I have uploaded the reply in file form.

Round 2

Reviewer 1 Report

recommend acceptance in the present form 

Author Response

I have revised the article.

Reviewer 2 Report

The revised version of polymers-2261812 has improved the Introduction and bibliographic part.

Author Response

I have put your reply in the file, please check it.

Round 3

Reviewer 2 Report

I consider the new revision version as largely improved and so, I recommend iy for publication.